

# The Gauge-Higgs legacy of the LHC run II

Anke Biekötter[1], Tyler Corbett[2] and Tilman Plehn[1]

**1** Institut für Theoretische Physik, Universität Heidelberg, Germany
**2** Niels Bohr International Academy and Discovery Centre,
Niels Bohr Institute, University of Copenhagen, Denmark

⋆ biekoetter@thphys.uni-heidelberg.de

## Abstract

We present a global analysis of the Higgs and electroweak sector based on LHC Run II and electroweak precision observables. We show which measurements provide the leading constraints on Higgs-related operators, and how the achieved LHC precision makes it necessary to combine rate measurements with electroweak precision observables. The SFitter framework allows us to include kinematic distributions beyond pre-defined AT-LAS and CMS observables, independently study correlations, and avoid Gaussian assumptions for theory uncertainties. These Run II results are a step towards a precision physics program at the LHC, interpreted in terms of effective operators.



## Content

# 1   Introduction

After the discovery of a light, likely fundamental Higgs boson largely compatible with the Standard Model [1–3], the LHC has focused on precision studies of electroweak symmetry breaking [4,5]. From a theoretical as well as from an experimental perspective, the appropriate interpretation framework for such LHC precision analyses are effective Lagrangians [6–16]. They require us to fix the (propagating) particle content and the underlying symmetry structure. For the former, experimental observations point to the Standard Model content, possibly extended by a dark matter agent coupling to the Higgs sector. Concerning the interactions, we can assume the Higgs doublet structure of the Standard Model, which intertwines the Higgs sector and the electroweak gauge sector [17–20]. The corresponding analyses based on Run I data [21–39] and first analyses based on Run II data [40,41] prove that the LHC has successfully transitioned to a precision physics experiment.

In the effective theory version of the Standard Model [16] we assume that departures of Higgs or gauge boson interactions from their SM predictions are characterized by a new energy scale $\Lambda$. It is crucial that this energy scale is not kinematically accessible at the LHC, which means that the corresponding new particles never appear on their mass shell. This condition defines the validity of the EFT approach [42,43]. Because the range of energies accessible in the kinematic regime of the LHC does not guarantee a strong hierarchy of scales [44], we can then think of an effective Lagrangian representing classes of new physics models [45–52].

One of the great advantages of the SMEFT framework is that it allows for global analyses of LHC measurements not only in the Higgs and electroweak gauge sectors, but also in the QCD sector [53–59], the top sector [60–65], or the flavor sector [66]. For LHC Run I there exist analyses combining Higgs measurements with LEP data [67–71] or, even better, di-boson production at the LHC searching for anomalous triple gauge vertices [23,72–77]. At this point we find that in the effective Lagrangian framework the LHC limits are surpassing the LEP limits, because effective operators with a momentum dependence can be tested either through high precision or through large momentum flow [78,79]. Similarly, at the level of Run II precision we should not hard-code the electroweak precision constraints into our operator basis [80–85]. Fermionic operators affect electroweak precision data and LHC data in different combinations with the usual bosonic operators, and this correlation generally weakens the constraints on operators contributing to Higgs physics only. This brings the number of SMEFT operators considered in our global Higgs analysis to 20, plus invisible decays. Two of these operators turn out to be successfully constrained by non-Higgs observables, so they do not have to be considered in the actual analysis.

In this paper we present an SFITTER analysis of the Higgs and gauge sector at the LHC and electroweak precision data. As usual, we do not rely on pre-defined results from ATLAS and CMS, but evaluate event counts in total rate measurements and kinematic distributions using our in-house framework whenever available [86–88]. This allows us to correlate systematic uncertainties, define our own treatment of theoretical uncertainties, and account for non-Gaussian constraints. We start by defining our relevant operator basis in Sec. 2 and 3. We then compare possible Higgs-sector constraints on operators measured in other LHC analyses in Sec. 4. With this operator basis we then report on a global LHC analysis, starting with a comparison of Run I and Run II results, adding electroweak precision observables, and discussing the interplay of the two kinds of operators in detail in Sec. 6. Our final result brings us a significant step closer to a global SFITTER SMEFT analysis.

## 2 Higgs and gauge sector

The linear effective Lagrangian is an $SU(3)_c \otimes SU(2)_L \otimes U(1)_Y$-symmetric extension of the renormalizable Standard model, but with the SM field content. It is ordered by inverse powers of the new physics scale [6–13, 17–20],

$$\mathcal{L} = \sum_x \frac{f_x}{\Lambda^2} \mathcal{O}_x .$$  (1)

Neglecting lepton number violation at dimension five the first order of new physics effects is dimension six, with 59 baryon-number conserving operators, barring flavor structure and Hermitian conjugation [9–13]. We follow the definition of the relevant operator basis of Ref. [24,25]: first, we restrict the initial set to *P*-even and *C*–even operators*. We then use the equations of motion to rotate to a basis where there are no blind directions linked to electroweak precision data. We then neglect all operators that cannot be studied at the LHC yet or which are strongly constrained from other LHC measurements. This includes the *HHH* vertex [91–96], the Higgs interactions with light-generation fermions, and some operators discussed in Sec. 4. We are left with 18 dimension-6 operators, ten of which do not influence electroweak precision observables at tree level [24, 25],

$$\mathcal{O}_{GG} = \phi^\dagger \phi \, G_{\mu\nu}^a G^{a\mu\nu} \qquad \mathcal{O}_{WW} = \phi^\dagger \hat{W}_{\mu\nu} \hat{W}^{\mu\nu} \phi \qquad \mathcal{O}_{BB} = \phi^\dagger \hat{B}_{\mu\nu} \hat{B}^{\mu\nu} \phi$$

$$\mathcal{O}_W = (D_\mu \phi)^\dagger \hat{W}^{\mu\nu} (D_\nu \phi) \qquad \mathcal{O}_B = (D_\mu \phi)^\dagger \hat{B}^{\mu\nu} (D_\nu \phi)$$

$$\mathcal{O}_{\phi 2} = \frac{1}{2} \partial^\mu (\phi^\dagger \phi) \partial_\mu (\phi^\dagger \phi) \quad \mathcal{O}_{WWW} = \text{Tr}\left( \hat{W}_{\mu\nu} \hat{W}^{\nu\rho} \hat{W}_\rho^\mu \right)$$  (2)

$$\mathcal{O}_{e\phi,33} = \phi^\dagger \phi \, \bar{L}_3 \phi e_{R,3} \qquad \mathcal{O}_{u\phi,33} = \phi^\dagger \phi \, \bar{Q}_3 \tilde{\phi} u_{R,3} \qquad \mathcal{O}_{d\phi,33} = \phi^\dagger \phi \, \bar{Q}_3 \phi d_{R,3} .$$

The covariant derivative acting on the Higgs is $D_\mu = \partial_\mu + ig' B_\mu/2 + ig\sigma_a W_\mu^a/2$, and the field strengths are $\hat{B}_{\mu\nu} = ig' B_{\mu\nu}/2$ and $\hat{W}_{\mu\nu} = ig\sigma^a W_{\mu\nu}^a/2$. This ad-hoc rescaling of the field strength can be motivated through our expectations from known UV-completions, but it has no effect on our analysis or its interpretation. The effective Lagrangian which we use to interpret Higgs and triple-gauge vertex (TGV) measurements at the LHC is

$$\mathcal{L}_{\text{eff}} \supset -\frac{\alpha_s}{8\pi} \frac{f_{GG}}{\Lambda^2} \mathcal{O}_{GG} + \frac{f_{WW}}{\Lambda^2} \mathcal{O}_{WW} + \frac{f_{BB}}{\Lambda^2} \mathcal{O}_{BB}$$

$$+ \frac{f_W}{\Lambda^2} \mathcal{O}_W + \frac{f_B}{\Lambda^2} \mathcal{O}_B + \frac{f_{\phi 2}}{\Lambda^2} \mathcal{O}_{\phi 2} + \frac{f_{WWW}}{\Lambda^2} \mathcal{O}_{WWW}$$

$$+ \frac{f_\tau m_\tau}{v\Lambda^2} \mathcal{O}_{e\phi,33} + \frac{f_b m_b}{v\Lambda^2} \mathcal{O}_{d\phi,33} + \frac{f_t m_t}{v\Lambda^2} \mathcal{O}_{u\phi,33} + \text{invisible decays} .$$  (3)

For invisible Higgs decays we do not include a term in the Lagrangian and consequently describe it in terms of an invisible partial width. It is best constrained through WBF Higgs production [97–99]. All operators except for $\mathcal{O}_{WWW}$ contribute to Higgs interactions. Their contributions to the several Higgs vertices, including non-SM Lorentz structures, are described in Ref. [21, 22, 100].

Some of the operators in Eq. (2) contribute to the self-interactions of the electroweak gauge bosons. They can be linked to specific deviations in the Lorentz structures entering the *WWZ* and *WWγ* interactions, historically written as $\kappa_\gamma, \kappa_Z, g_1^Z, g_1^\gamma, \lambda_\gamma,$ and $\lambda_Z$ [101]. After using

---

*Before trying to prove for example *CP*-violation through a global fit we advocate dedicated *CP* tests for the Higgs and gauge sector [89, 90].

electromagnetic gauge invariance to fix $g_1^\gamma = 1$, the shifts are defined by

$$
\begin{aligned}
\Delta\mathcal{L}_{\text{TGV}} =& -ie\,(\kappa_\gamma - 1)\,W_\mu^+ W_\nu^- \gamma^{\mu\nu} - \frac{ie\lambda_\gamma}{m_W^2}\,W_{\mu\nu}^+ W^{-\nu\rho} \gamma_\rho^{\ \mu} - \frac{ig_Z \lambda_Z}{m_W^2}\,W_{\mu\nu}^+ W^{-\nu\rho} Z_\rho^{\ \mu} \\
& -ig_Z\,(\kappa_Z - 1)\,W_\mu^+ W_\nu^- Z^{\mu\nu} - ig_Z\,(g_1^Z - 1)\,\left(W_{\mu\nu}^+ W^{-\mu} Z^\nu - W_\mu^+ Z_\nu W^{-\mu\nu}\right) \\
=& -ie\,\frac{g^2 v^2}{8\Lambda^2}\,(f_W + f_B)\,W_\mu^+ W_\nu^- \gamma^{\mu\nu} - ie\,\frac{3g^2 f_{WWW}}{4\Lambda^2}\,W_{\mu\nu}^+ W^{-\nu\rho} \gamma_\rho^{\ \mu} \\
& -ig_Z\,\frac{g^2 v^2}{8c_w^2 \Lambda^2}\,\left(c_w^2 f_W - s_w^2 f_B\right)\,W_\mu^+ W_\nu^- Z^{\mu\nu} - ig_Z\,\frac{3g^2 f_{WWW}}{4\Lambda^2}\,W_{\mu\nu}^+ W^{-\nu\rho} Z_\rho^{\ \mu} \\
& -ig_Z\,\frac{g^2 v^2 f_W}{8c_w^2 \Lambda^2}\,\left(W_{\mu\nu}^+ W^{-\mu} Z^\nu - W_\mu^+ Z_\nu W^{-\mu\nu}\right),
\end{aligned}
\tag{4}
$$

where $e = g s_w$ and $g_Z = g c_w$. The two notational conventions are equivalent for gauge-invariant models and linked as

$$
\kappa_\gamma = 1 + \frac{g^2 v^2}{8\Lambda^2}(f_W + f_B), \qquad\qquad \kappa_Z = 1 + \frac{g^2 v^2}{8c_w^2 \Lambda^2}\left(c_w^2 f_W - s_w^2 f_B\right),
$$

$$
g_1^Z = 1 + \frac{g^2 v^2}{8c_w^2 \Lambda^2} f_W, \qquad\qquad g_1^\gamma = 1, \qquad\qquad \lambda_\gamma = \lambda_Z = \frac{3g^2 m_W^2}{2\Lambda^2} f_{WWW} .
\tag{5}
$$

The three Wilson coefficients relevant for our analysis of di-boson production are $f_B$, $f_W$ and $f_{WWW}$, plus the operators influencing electroweak precision data discussed in Section 3. To get a very rough idea what kind of new physics scales we can probe in the electroweak gauge and Higgs sector we quote the typical range from the global Run I analyses,

$$
\frac{\Lambda}{\sqrt{|f|}} \gtrsim 300 \ldots 500 \text{ GeV} \qquad \text{(Higgs-gauge analysis at Run I [23]).}
\tag{6}
$$

We note that already the Run I di-boson measurements clearly outperform the corresponding LEP measurements evaluated in the effective operator basis of Eq.(3).

If we deviate from this scenario and consider instead the more generic non-linear or chiral effective Lagrangian [102–105], the parametrization would be extended. In the most generic scenario, the TGV couplings defined above depend on a larger number of parameters and the correlations from gauge dependence are lost. Furthermore, the deviations generated by non-linear operators in the TGVs could be completely de-correlated to the deviations generated in the Higgs interactions. For the Higgs sector alone, the linear and non-linear analyses can be trivially mapped onto each other [21, 22].

## 3 Electroweak precision sector

While the Lagrangian in Eq.(3) does not include tree-level contributions to electroweak precision observables, we know that at the level of 13 TeV data the corresponding operators should not be neglected [40, 41, 80–85]. This means that we need to add two bosonic operators

$$
\mathcal{O}_{\phi 1} = (D_\mu \phi)^\dagger \phi \phi^\dagger (D^\mu \phi), \qquad\qquad \mathcal{O}_{BW} = \phi^\dagger \hat{B}_{\mu\nu} \hat{W}^{\mu\nu} \phi ,
\tag{7}
$$

Table 1: List of operators affecting electroweak precision observables and their effect on fermionic couplings testable at the LHC.

| operator | $H f \bar{f}$ | $Z q q$ | $W q q'$ | $Z l \bar{l}$ | $W l \nu$ |
|---|---|---|---|---|---|
| $\mathcal{O}_{\phi 1}$ | × | × | × | × | × |
| $\mathcal{O}_{BW}$ | | × | × | × | × |
| $\mathcal{O}_{\phi Q}^{(3)}$ | | × | × | | |
| $\mathcal{O}_{\phi Q}^{(1)}$ | | × | | | |
| $\mathcal{O}_{\phi u}^{(1)}$ | | × | | | |
| $\mathcal{O}_{\phi d}^{(1)}$ | | × | | | |
| $\mathcal{O}_{\phi e}^{(1)}$ | | | | × | |

which affect gauge and Higgs interactions. In addition we consider the fermionic Higgs-gauge operators

$$
\mathcal{O}_{\phi Q,ij}^{(1)} = \phi^\dagger (i \overleftrightarrow{D}_\mu \phi)(\bar{Q}_i \gamma^\mu Q_i), \qquad \mathcal{O}_{\phi Q,ij}^{(3)} = \phi^\dagger (i \overleftrightarrow{D}_\mu^a \phi)(\bar{Q}_i \gamma^\mu \frac{\sigma_a}{2} Q_i),
$$

$$
\mathcal{O}_{\phi L,ij}^{(1)} = \phi^\dagger (i \overleftrightarrow{D}_\mu \phi)(\bar{L}_i \gamma^\mu L_i), \qquad \mathcal{O}_{\phi L,ij}^{(3)} = \phi^\dagger (i \overleftrightarrow{D}_\mu^a \phi)(\bar{L}_i \gamma^\mu \frac{\sigma_a}{2} L_i),
$$

$$
\mathcal{O}_{\phi u,ij}^{(1)} = \phi^\dagger (i \overleftrightarrow{D}_\mu \phi)(\bar{u}_{R,i} \gamma^\mu u_{R,i}), \qquad \mathcal{O}_{LLLL} = (\bar{L}_1 \gamma_\mu L_2)(\bar{L}_2 \gamma^\mu L_1),
$$

$$
\mathcal{O}_{\phi d,ij}^{(1)} = \phi^\dagger (i \overleftrightarrow{D}_\mu \phi)(\bar{d}_{R,i} \gamma^\mu d_{R,i}),
$$

$$
\mathcal{O}_{\phi e,ij}^{(1)} = \phi^\dagger (i \overleftrightarrow{D}_\mu \phi)(\bar{e}_{R,i} \gamma^\mu e_{R,i}),
$$

$$
\mathcal{O}_{\phi ud,ij}^{(1)} = \tilde{\phi}^\dagger (i \overleftrightarrow{D}_\mu \phi)(\bar{u}_{R,i} \gamma^\mu d_{R,i}). \tag{8}
$$

The operator $\mathcal{O}_{\phi ud,ij}^{(1)}$ contains the charged current $\bar{u}_R \gamma^\mu d_R$ [41,106–108]. Given that it does not interfere with the Standard Model and the known flavor physics constraints we will ignore it in our analysis, the same way we exclude for example dipole operators.

The first eight operators generate anomalous weak boson couplings to fermions, while they do not affect the Higgs coupling to fermions, see Tab. 1. They do modify the Higgs couplings to weak bosons and fermions, for instance introducing point-like $HVff$ interactions. We also include the 4-lepton operator $\mathcal{O}_{LLLL}$ as it induces a shift in the Fermi constant. For our study we assume diagonal and generation independent Wilson coefficients for the fermionic operators affecting the electroweak currents. Further, we will eliminate the leptonic operators $\mathcal{O}_{\phi L,ii}^{(1)}$ and $\mathcal{O}_{\phi L,ii}^{(3)}$ using the equations of motion:

$$
2\mathcal{O}_{\phi 2} + 2\mathcal{O}_{\phi 4} = \sum_{ij}\left( y_{ij}^e (\mathcal{O}_{e\phi,ij})^\dagger + y_{ij}^u \mathcal{O}_{u\phi,ij} + y_{ij}^d (\mathcal{O}_{d\phi,ij})^\dagger + \text{h.c.}\right) - \frac{\partial V(h)}{\partial h},
$$

$$
2\mathcal{O}_B + \mathcal{O}_{BW} + \mathcal{O}_{WW} + g^2\left(\mathcal{O}_{\phi 4} - \frac{1}{2}\mathcal{O}_{\phi 2}\right) = -\frac{g^2}{4}\sum_i\left(\mathcal{O}_{\phi L,ii}^{(3)} + \mathcal{O}_{\phi Q,ii}^{(3)}\right),
$$

$$
2\mathcal{O}_B + \mathcal{O}_{BW} + \mathcal{O}_{BB} + g'^2\left(\mathcal{O}_{\phi 1} - \frac{1}{2}\mathcal{O}_{\phi 2}\right) =
$$

$$
-\frac{g'^2}{2}\sum_i\left(-\frac{1}{2}\mathcal{O}_{\phi L,ii}^{(1)} + \frac{1}{6}\mathcal{O}_{\phi Q,ii}^{(1)} - \mathcal{O}_{\phi e,ii}^{(1)} + \frac{2}{3}\mathcal{O}_{\phi u,ii}^{(1)} - \frac{1}{3}\mathcal{O}_{\phi d,ii}^{(1)}\right). \tag{9}
$$

Assuming a universal flavor structure this leaves us with the additional contributions to our effective Lagrangian,

$$\mathcal{L}_{\text{eff}} \supset + \frac{f_{\phi 1}}{\Lambda^2}\mathcal{O}_{\phi 1} + \frac{f_{BW}}{\Lambda^2}\mathcal{O}_{BW} + \frac{f_{LLLL}}{\Lambda^2}\mathcal{O}_{LLLL}$$
$$+ \frac{f_{\phi Q}^{(1)}}{\Lambda^2}\mathcal{O}_{\phi Q}^{(1)} + \frac{f_{\phi d}^{(1)}}{\Lambda^2}\mathcal{O}_{\phi d}^{(1)} + \frac{f_{\phi u}^{(1)}}{\Lambda^2}\mathcal{O}_{\phi u}^{(1)} + \frac{f_{\phi e}^{(1)}}{\Lambda^2}\mathcal{O}_{\phi e}^{(1)} + \frac{f_{\phi Q}^{(3)}}{\Lambda^2}\mathcal{O}_{\phi Q}^{(3)} \,. \tag{10}$$

Together with Eq.(3) this defines the operator basis for our global analysis, altogether 18 operators plus the invisible Higgs branching ratio. While the additional operators affect many of our LHC measurements, they are also strongly constrained by electroweak precision observables. The challenge is that the bosonic operators in Eq.(7) and the fermionic operators in Eq.(8) not only contribute to electroweak precision physics, but also to di-boson or Higgs production at an observable level, where they are included e.g. in our study of triple gauge vertices. Because the two data sets combine very different combinations of operators, we have to combine our Run II analysis with a set of electroweak precision observables. We follow Ref.[81] and review this approach briefly. Our $Z$-pole observables are

$$\left\{ \Gamma_Z, \sigma_h^0, \mathcal{A}_l(\tau^{\text{pol}}), R_l^0, \mathcal{A}_l(\text{SLD}), A_{\text{FB}}^{0,l}, R_c^0, R_b^0, \mathcal{A}_c, \mathcal{A}_b, A_{\text{FB}}^{0,c}, A_{\text{FB}}^{0,b}(\text{SLD/LEP-I}) \right\}, \tag{11}$$

with measurements and correlations taken from Ref [109]. We also include the $W$-observables

$$\left\{ m_W, \Gamma_W, \text{BR}(W \to l \nu) \right\}, \tag{12}$$

with values taken from Ref. [110]. The SM predictions for these observables are taken from Ref. [111]. We note that for the SM prediction of the $W$-mass this includes the full one- and two-loop EW and two-loop QCD corrections of $\mathcal{O}(\alpha\alpha_s)$ as well as some 3-loop contributions. The contributions of our dimension-6 operators can be found in Ref. [81], where we limit ourselves to linear contributions from the higher-dimensional operators considered in our fit. This approximation is justified as long as the dimension-6 corrections are small, *i.e.* $f m_Z^2/\Lambda^2 \ll 1$ assuming that the typical energy scale of electroweak precision data is around $m_Z$. The standard analyses of electroweak precision data indeed give individual limits of the kind

$$\frac{\Lambda}{\sqrt{|f|}} \gtrsim 4 \dots 10 \text{ TeV} \qquad \text{(electroweak precision data [111]).} \tag{13}$$

These limits significantly exceeds the expected sensitivity of the global LHC analysis from Eq.(6), which naively suggests that it is not necessary to combine the two sectors. In the discussion of our global fit in Section 6 we will see how the fermionic Higgs-gauge operators nevertheless lead to visible effects at the LHC.

# 4 QCD and top sectors

An operator which should be added to any basis confronted with LHC data is the anomalous triple gluon coupling

$$\mathcal{O}_G = f_{abc} G_{a\nu}^\rho G_{b\lambda}^\nu G_{c\rho}^\lambda, \qquad \text{with} \qquad \mathcal{L}_{\text{eff}} \supset \frac{g_s f_G}{\Lambda^2}\mathcal{O}_G \,, \tag{14}$$

with $G_a^{\rho\nu} = \partial^\rho G_a^\nu - \partial^\nu G_a^\rho - i g_s f_{abc} G^{b\rho} G^{c\nu}$. It contributes to any gluon-induced LHC process, for instance Higgs production with a hard jet. While it only affects kinematic distributions

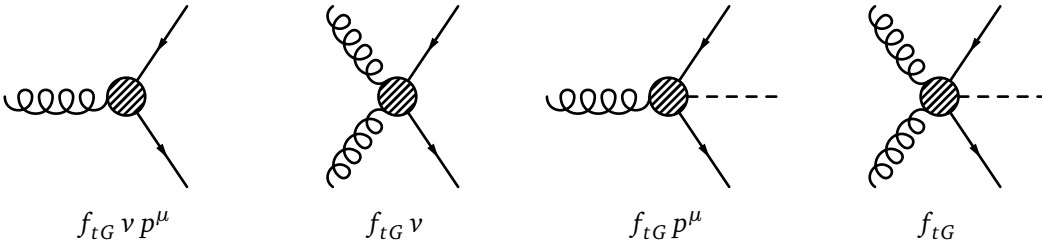

Figure 1: Interactions through the chromo-magnetic top operator. The vertices scaling with $p^\mu$ come from the derivative in the field strength, while those scaling with $\nu$ are generated by the commutator component of the field strength.

with an additional hard parton, it needs to be taken into account when we use the same distribution to separate $\mathcal{O}_{u\phi,33}$ effects from $\mathcal{O}_{GG}$. On the other hand, it can be constrained by ATLAS multi-jet data at 13 TeV, giving the 95% CL limits [59]

$$\frac{\Lambda}{\sqrt{f_G}} > 5.2\,(5.8)\,\text{TeV} \qquad \text{observed (expected) from multi-jets.} \tag{15}$$

This limits the possible effects on Higgs production rates beyond anything a global Higgs analysis would be sensitive to in the absence of a dedicated enhancement mechanism in Higgs rates.

A critical feature of Higgs analyses is the combination of direct and indirect measurements of the top-Higgs coupling in gluon fusion and associated Higgs-top production [112,113]. The chromo-magnetic top operator

$$\mathcal{O}_{tG} = (\bar{Q}\sigma^{\mu\nu}T^A u_R)\,\tilde{H}\,G^A_{\mu\nu}\,, \tag{16}$$

will, in principle, affect these observables [40] and has been studied extensively in top-EFT analyses [65]. The interaction vertices induced by $\mathcal{O}_{tG}$ are shown in Fig. 1. The first two diagrams contribute to top pair production, the second set to $t\bar{t}H$ production. In each case one of the interactions is proportional to the momentum flowing through the vertex.

To constrain $f_{tG}$ in a Higgs fit we can consider gluon fusion and $t\bar{t}H$ production with additional jets. However, extra hard gluons in the final state are a typical higher-order effect and likely suppressed. Alternatively, we can use momentum-dependent distributions in $t\bar{t}H$ production. The third vertex in Fig. 1 appears to allow for such effects as it only includes a single gluon, however this momentum dependence as well as the triple gluon vertex of the SM will be compensated by an additional propagator in the amplitude resulting in no additional growth with momentum. This lack of growth with momentum is demonstrated in Figure 2 below which shows the shape of the $H_T$ distribution does not change dramatically with increasing $f_{tG}$.

We can estimate the extent to which this operator can be constrained. The most promising distribution currently available is the $H_T$ distribution in the all-hadronic

$$pp \to t\bar{t}H \to t\bar{t}\,b\bar{b} \tag{17}$$

signature released by CMS [114]. In Fig. 2 we reproduce their $H_T$ distribution as well as the distribution in the presence of two benchmark values of $f_{tG}$. We generate the relevant $ttH$ process merged with one additional jet using MADGRAPH5 [115] and PYTHIA8 [116], combined

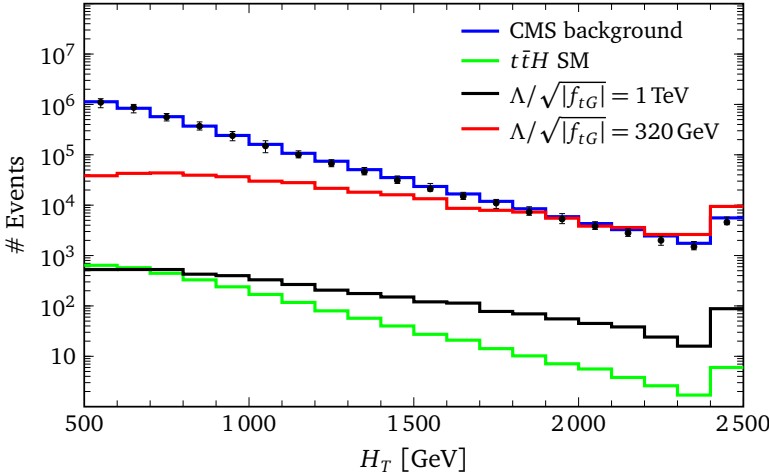

Figure 2: $H_T$ distributions for $t\bar{t}H$ production for the Standard Model, $\Lambda/\sqrt{|f_{tG}|} = 1$ TeV corresponding to the top physics limit, and $\Lambda/\sqrt{|f_{tG}|} = 320$ GeV corresponding to the Higgs physics limit. The background estimate and the data points are from Ref. [114].

with DELPHES3 [117]. The two benchmarks each correspond to

$$\frac{\Lambda}{\sqrt{|f_{tG}|}} \gtrsim 1 \text{ TeV} \qquad\qquad \text{(top sector [65]),}$$

$$\frac{\Lambda}{\sqrt{|f_{tG}|}} \gtrsim 320 \text{ GeV} \qquad\qquad \text{(Higgs sector [40]).} \qquad (18)$$

From Fig. 2 we see that our expected sensitivity is comparable to the Higgs study and not competitive with the top-sector constraints. This comes as no surprise: the $t\bar{t}H$ cross section is phase space suppressed relative to $t\bar{t}$ production and its cross section at the LHC is measured to be approximately three orders of magnitude below that of $t\bar{t}$ production [118, 119]. In addition, $t\bar{t}H$ production is plagued by large backgrounds. This implies statistical limitations on measuring $f_{tG}$ in $t\bar{t}H$, so indeed $\mathcal{O}_{tG}$ and $\mathcal{O}_G$ can both be neglected in global Higgs analyses in the near future. There are projections, however, that for the 14 TeV LHC with a luminosity of 3/ab that constraints on $\mathcal{O}_{tG}$ from $t\bar{t}H$ will exceed those of $t\bar{t}$ [112].

## 5 SFitter framework

In SFITTER analyses we prefer not to rely on the pre-processed rate modifiers by ATLAS and CMS whenever possible. Instead, we extract the signal and background rates from the experimental publications and apply our own uncertainty treatment. This includes correlated and uncorrelated systematic uncertainties as well as a flat likelihood within the allowed band by theoretical uncertainties. For analyses using multivariate analysis techniques, where the number of events in each signal region is only illustrated after simple cuts rather than the full analysis, we implement the signal strength modifiers but separate for example the theory uncertainties. All signal efficiencies and higher-order effects we extract as the difference between our simulation and the numbers quoted by ATLAS and CMS.

For Higgs and di-boson signals we use MADGRAPH5 [115] for the event generation, PYTHIA6 [116] for parton shower and hadronization, and DELPHES3 [117] for the detector simulation. Branching ratios including dimension-6 effects are given by the extended version

Table 2: List of Run II Higgs measurements included in our analysis. For the $m_{Vh}$ distribution our highest-momentum bin with observed events starts at $m_{Vh} = 990$ GeV and 1.2 TeV for the $0\ell$ and $1\ell$ final states.

| production | decay | | ATLAS | CMS |
|---|---|---|---|---|
| | $h \to WW$ | | [120, 121] | [122–124] |
| | $h \to ZZ$ | | [121, 125] | [123, 124, 126, 127] |
| | $h \to \gamma\gamma$ | | [128] | [129] |
| | $h \to \tau\tau$ | | [121] | [123, 124, 130] |
| | $h \to Z\gamma$ | | [131] | [132] |
| WBF | $h \to \text{inv}$ | | | [133] |
| WBF | $h \to \tau\tau$ | | | [130] |
| $Vh$ | $h \to b\bar{b}$ | | [134] | [135] |
| $Vh$ | $h \to \tau\tau$ | | | [136] |
| $Vh$ | $h \to \text{inv}$ | | [137] | [138] |
| $Vh$ | $h \to b\bar{b}$ ($m_{Vh}$) | | [139] | |
| $t\bar{t}h$ | $h \to \gamma\gamma$ | | [118] | [129] |
| $t\bar{t}h$ | $h \to ZZ \to 4\ell$ | | [118] | [126, 127] |
| $t\bar{t}h$ | $h \to WW, ZZ, \tau\tau$ | | [121] | [123, 124] |
| $t\bar{t}h$ | $h \to b\bar{b}$ | | [140] | [141] |

of HDECAY [142]. For new physics effects in the production process we use the same tool chain as for the Standard Model, combined with our FEYNRULES [143] implementation of the dimension-6 operators and assume that detector effects as well as higher-order corrections scale with the SM case in the fiducial volume of the SM-like measurement. For total rate measurements using the bulk of the phase space this approximation is obviously justified. For our kinematic distributions this is less clear, so we have checked that our approach is approximately correct [80–85, 144]. Corrections to diboson production have been calculated and should eventually be included [145].

As usual for our SFITTER analysis we allow for the modification of the production amplitude through dimension-6 operators including the interference with the SM amplitude and the squared term in the Wilson coefficient. The latter becomes relevant whenever the interference with the Standard Model is suppressed. Given the estimates of Eq.(6) and Eq.(13) we simplify our analysis by neglecting diagrams which are modified by bosonic and fermionic operators at the same time and interfere with the SM amplitude. In our discussion of the results we will see that indeed large effects from the fermionic operators do not appear in this topology. Finally, we neglect dimension-6 squared contributions of the fermionic operators to the gauge boson branching ratios, because they will be strongly suppressed following Eq.(13) with a typical energy scale $m_V$ in the gauge boson decays. For the same reason we neglect the effects of the fermionic operators on the decays of gauge bosons coming from Higgs decays. The hierarchy of scales combined with the well-defined external energy scale $E \lesssim m_H$ will render them numerically irrelevant.

For Higgs and di-boson we start with the set of Run I measurements discussed in Refs [21–23]. We add the Run II Higgs measurements shown in Tab. 2 and the Run II di-boson measurements shown in Tab. 3. Because the dimension-6 Lagrangian introduces new Lorentz structures and hence predicts significantly different event kinematics from the Standard Model, kinematic

distributions scaling with energy are especially powerful. An attractive case is a $m_{VH}$ distribution from an ATLAS resonance search [139], which we include for the zero-lepton and one-lepton final states. We re-bin the reported result such that the most relevant high bins include a statistically meaningful number of events, giving us measurements exceeding $m_{VH} = 1$ TeV.[†] The other side of the kinematics medal is that differential measurements from $H \to 4\ell$ decays can be safely neglected in a global analysis. The reason is that the momentum flow through the Higgs decay vertex is cut off by the on-shell condition, so any measurement in $VH$ or WBF production will surpass their impact on a global analysis [146].

Based on all measurements we first construct a multi-dimensional, full exclusive likelihood map. As long as we are only interested in small deviations from the Standard Model, a key assumption to be able to use an effective field theory approach, we can assume that local SM-like minima are also the global minima in this likelihood map. There exist three standard ways to explore the log-likelihood distribution around the minimum: first, we can use a naive, MINUIT-like approach, approximating the functional form around the minimum by a quadratic function. This assumption is not appropriate once we allow for non-Gaussian errors, for example a flat shape covering the theoretical error bar. Second, we can construct a Markov chain over the parameter space. Here the problem is that different directions in the space of Wilson coefficients behave differently, which makes it hard to define a universal and efficient proposal function. Nevertheless, we check our results against such a Markov chain analysis and usually find encouraging agreement. For our numerical analysis we define 10.000 toy measurements, modeling the Poisson, Gaussian or flat input distributions. For each toy experiment we determine the best-fitting point in the space of Wilson coefficients, combine these values to a histogram, effectively profile over the remaining parameters, and determine the 68% and 95% ranges around the SM-like central value. For the error bands we require the log-likelihood values at the lower and upper ends to be identical.

Because our approach gives us full control over the log-likelihood distribution we can compare these limits with a dual Gaussian fit to the log-likelihood in one dimension. We find good agreement for all Wilson coefficients, even though Fig. 3 shows that for example the profile likelihood for $f_W$ does not have a symmetric Gaussian shape. Obviously, the shape for the invisible Higgs width is distorted, because it does not allow for negative branching ratios. While we quote the error bars for the non-Gaussian analysis we quote the results from the Gaussian fit whenever we give a best-fit point for a Wilson coefficient. For additional details on the SFITTER framework we refer to Refs. [86–88].

Table 3: List of Run I and Run II di-boson measurements included in our analysis. The maximum value in GeV indicates the lower end of the highest-momentum bin we consider.

|  | channel | distribution | #bins | max [GeV] |  |
|---|---|---|---|---|---|
| 8 TeV | $WW \to \ell^+\ell'^- + \not{E}_T \, (0j)$ | leading $p_{T,\ell}$ | 4 | 350 | 20.3 fb$^{-1}$ [147] |
|  | $WW \to \ell^+\ell^{(\prime)-} + \not{E}_T \, (0j)$ | $m_{\ell\ell^{(\prime)}}$ | 7 | 575 | 19.4 fb$^{-1}$ [148] |
|  | $WZ \to \ell^+\ell^-\ell^{(\prime)\pm}$ | $m_T^{WZ}$ | 6 | 450 | 20.3 fb$^{-1}$ [149] |
|  | $WZ \to \ell^+\ell^-\ell^{(\prime)\pm} + \not{E}_T$ | $p_T^{Z \to \ell\ell}$ | 8 | 350 | 19.6 fb$^{-1}$ [150] |
| 13 TeV | $WZ \to \ell^+\ell^-\ell^{(\prime)\pm}$ | $m_T^{WZ}$ | 7 | 675 | 36.1 fb$^{-1}$ [151] |

---

[†]We would happily thank ATLAS for help with this analysis result and we are grateful to the actual authors communicating with us. However, our EFT analysis was officially considered as no appropriate re-casting of a $VH$ resonance search, so there is nothing we can thank ATLAS for.

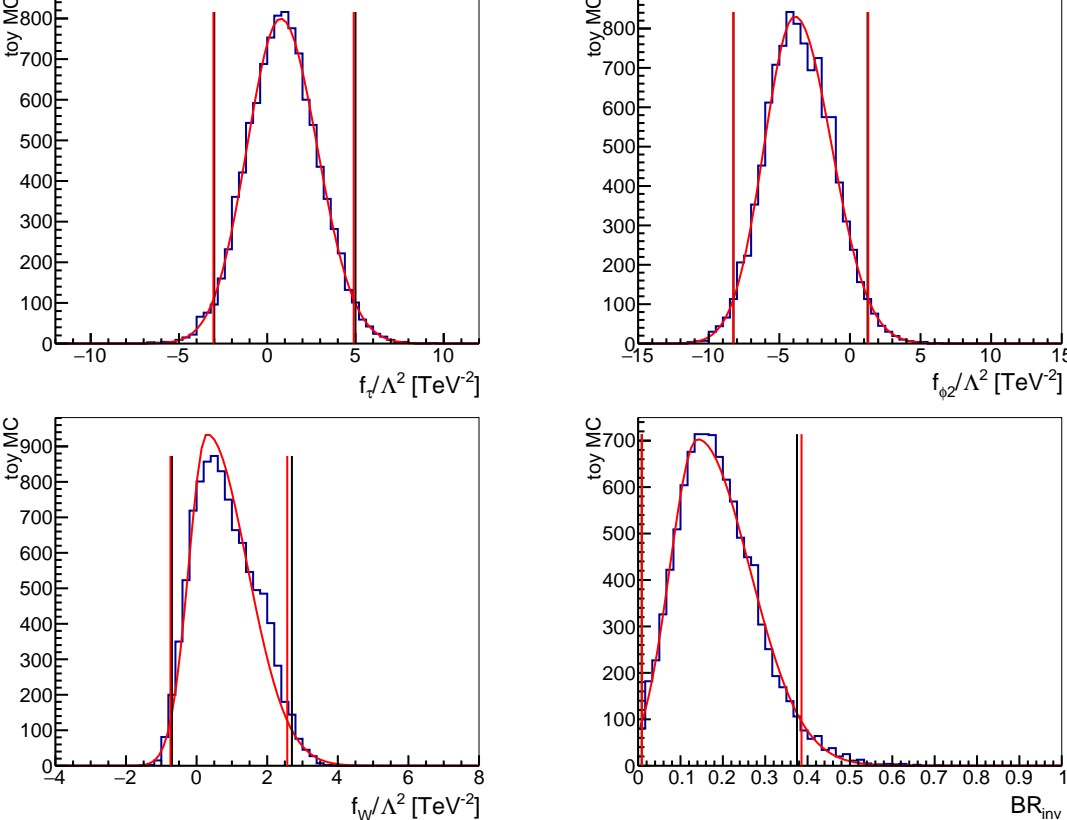

Figure 3: Distributions of the toy experiments for the operators $\mathcal{O}_\tau$, $\mathcal{O}_{\phi 2}$ and $\mathcal{O}_W$ as well as the invisible Higgs branching ratio, based on the full LHC data set. The lines show the 95% CL limits from the histogram (black) and the double-Gaussian fit (red).

One caveat applies to all analyses based on effective Lagrangians: we consider the dimension-6 Lagrangian of Eq.(3) and Eq.(10) the appropriate description of the physics effects beyond the Standard Model. Note that this statement by no means implies that for example the dimension-6-squared contributions have to be smaller than those from the dimension-6 interference with the Standard Model [152]. There exist many physics reasons why this could be a valid physics effect, and the discrepancy between the generic LHC reach given by Eq.(6) and the generic reach of electroweak precision data in Eq.(13) will be discussed as an example for such effects in the next section. Instead, we simply need to ensure that no particle of the UV-completions which we approximate with our effective Lagrangian contributes as a propagating degree of freedom on its mass shell [42,43]. To this end, computing the effects of dimension-8 operators can give useful hints about the validity of the dimension-6 truncation [153], but it does not have to.

Finally, in the spirit of the effective field theory we only consider SM-like scenarios, which means that we neglect all secondary solutions for example with switched signs of Yukawa couplings. Assuming weakly interacting new physics such effects require scales $\Lambda \sim m_H$, so we expect these models to be best tested in direct LHC searches rather than a global analysis. In any case, the observation of a sign switch for example in a Yukawa coupling as part of a global analysis would signal a breakdown of the renormalizable Standard Model and its symmetry structure and would prompt us to modify our SMEFT hypothesis. Of course, when it comes to searching for effects in kinematic distributions, these two search strategies are closely related, for example when we directly search for mass peaks in the same distributions that we indirectly

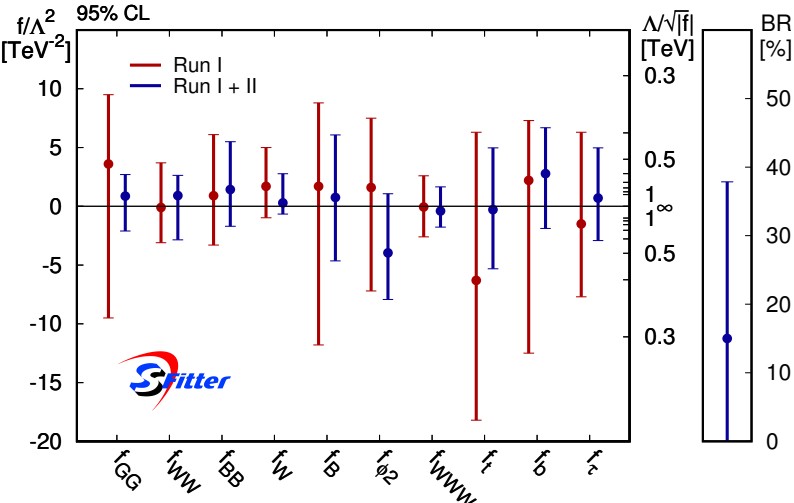

Figure 4: Allowed 95% CL ranges for individual Wilson coefficients $f_x/\Lambda^2$ from a one-dimensional profile likelihood. We show results from Run I (red) and using the additional Run II measurements (blue). We neglect all operators contributing to electroweak precision observables at tree level.

test for shoulders (as an early sign of a mass peak appearing in data) [42, 43].

## 6 Global analysis

Before we attempt a proper global analysis of the Higgs and electroweak gauge sector we can ask what the impact of the additional 13 TeV data given in Tabs. 2 and 3 is. Aside from a generic improvement in many of the standard measurements, we expect a significant impact from the new $t\bar{t}H$ measurements, the significant observation of fermionic Higgs decays, and from the re-casted $m_{VH}$ distribution to very large energies. In Fig. 4 we indeed see that the limits on $f_t$, $f_b$ and $f_\tau$ have improved by more than a factor of two. Obviously, the top Yukawa measurement directly affects the Higgs coupling to gluons, $\mathcal{O}_{GG}$, because it can only be extracted after we subtract the measured top loop contribution. Because $\mathcal{O}_{\phi 2}$ leads to a Higgs wave function renormalization and $\mathcal{O}_b$ modifies the total Higgs width, they are strongly correlated in the global analysis. After Run II they are not only well determined, both of them also show symmetric Gaussian log-likelihood distributions. We also see a very significant improvement in the limit on $f_W$ and $f_B$, which is driven by associated $VH$ production. However, from Fig. 3 we know that the error bar on $f_W$ is by no means symmetric and Gaussian due to the relative size of the linear and quadratic terms of the EFT, the parametrization of the theory prediction and further effects. The operators showing the least improvement compared to Run I are $\mathcal{O}_{WW}$ and $\mathcal{O}_{BB}$, reflecting the lack of high-impact kinematic WBF measurements in the Run II data set. Moreover, $\mathcal{O}_{WWW}$ only affects the gauge sector, and in Tab. 3 we see that the analysis is still dominated by a broad set of extremely successful kinematic measurements at Run I in view of a global gauge analysis.

Finally, our global limit on the Higgs branching ratio to invisible particles is

$$\text{BR}_{\text{inv}} < 38\% \qquad \text{at 95\% CL,} \tag{19}$$

with a best-fit point of $\text{BR}_{\text{inv}} = 14\%$. This is significantly weaker than the limits quoted for example by CMS [133], because our global analysis does not assume the underlying Higgs production rates to be SM-like. Indeed, we observe a strong correlation of the invisible branching

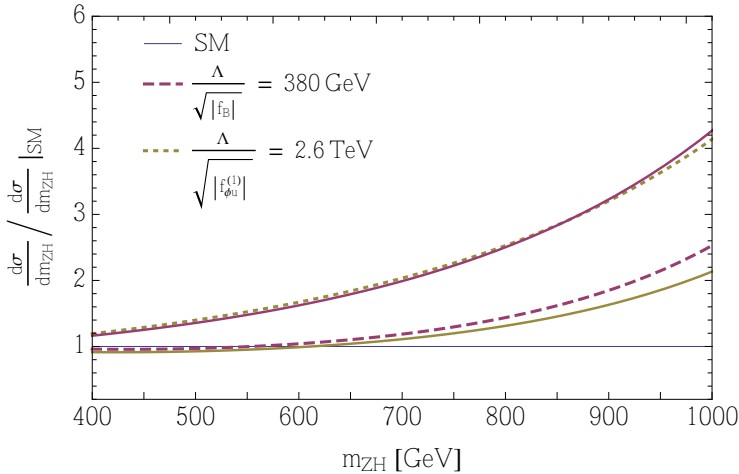

Figure 5: Invariant mass distribution $m_{ZH}$ normalized to the Standard Model. The dashed lines correspond to positive Wilson coefficients, while the solid lines correspond to the negative values of the Wilson coefficients with the same magnitude.

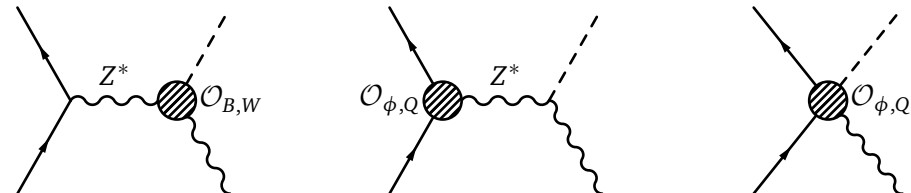

Figure 6: Dimension-6 contribution to $ZH$ production. We show sample diagrams for the usual bosonic corrections, the small fermionic corrections from a 3-point vertex, and the large fermionic corrections from a 4-point interaction.

ratio with $\mathcal{O}_{\phi 2}$ and its universal Higgs wave function renormalization. If rather than profiling over it we fix $f_{\phi 2} = 0$, our limit becomes $\text{BR}_{\text{inv}} < 26\%$ in agreement with the experimental results. Altogether, we find that Run II systematically probes energy scales $\Lambda/\sqrt{f}$ between 400 GeV and 800 GeV through Higgs measurement.

The large improvement of the limits on $\mathcal{O}_B$ at Run II forces us to consider the interplay with the fermionic operators from Eq.(10) and their limits from electroweak precision data, Eq.(13). From a scale separation point of view it is seems counter-intuitive that $\mathcal{O}_{\phi u}^{(1)}$ or $\mathcal{O}_{\phi Q}^{(3)}$, for which $\Lambda/\sqrt{f}$ is constrained around one order of magnitude more strongly than for $\mathcal{O}_W$ and much more strongly for all other operators shown in Fig. 4, should have any effect on the LHC analysis [80–85]. In Fig. 5 we see how the fermionic and bosonic operators affect for example $ZH$ production. The key observation is that the fermionic operator contributes via the 3-point $qqZ$ and the 4-point $qqHZ$ vertices, whereas the bosonic operators require the same $s$-channel $Z$-propagator we see in the Standard Model. We show the corresponding Feynman diagrams in Fig. 6. From the structure of the dimension-6 operator we can infer the scalings

$$\frac{g f_{\phi Q} v^2}{\Lambda^2} \quad (qqZ) \qquad \text{versus} \qquad \frac{g f_{\phi Q} v}{\Lambda^2} \quad (qqZH). \qquad (20)$$

The $m_{ZH}$ distribution shown in Fig. 5 is one of our most powerful observables. We have confirmed that for the fermionic operator it is entirely dominated by the 4-point interaction, even though the 3-point interaction does interfere with the Standard Model. This is due to the suppression of the amplitudes with propagating $Z$s due to the off-shell $Z$ which leads to a

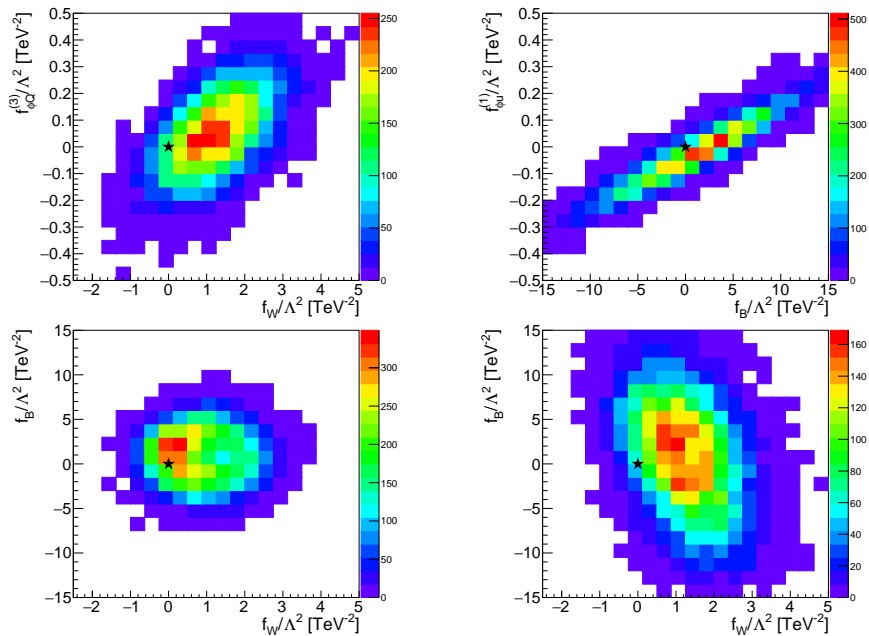

Figure 7: Correlations between the fermionic and bosonic operators (top row), and between the usual bosonic operators (bottom row). For the latter we show the purely LHC results (left) and the results after including the additional fermionic operators.

suppression going as $\sim 1/(m_{ZH}^2 - M_Z^2) < 1$ as well as the energy scaling in Eq.(20) as well as which will eventually also lead to unitarity violation [81].

It is interesting to see how two operators with an apparently very different new physics scale contribute to the $m_{ZH}$ distribution at around the same rate. This can be understood by the definitions of the operators which include a factor of the gauge coupling for each field strength tensor. While the 4-point contribution from $\mathcal{O}_{\phi u}^{(1)}$ lacks a second power of the gauge coupling $g'$ the definition of $\mathcal{O}_B$ adds two powers of the gauge coupling to the 3-point vertex. Over most of the parameter range shown in Fig. 5 the dimension-6-squared contribution dominates, giving us a mis-match of four powers of the coupling just from the definitions of the Wilson coefficients.

We confirm these findings in Fig. 7, where we show the resulting correlations in our global analysis, once we include the full Lagrangian of Eqs.(3) and (10). We see a clear correlation between $f_B$ and $f_{\phi u}^{(1)}$ from $ZH$ production, as well as between $f_W$ and $f_{\phi Q}^{(3)}$ from $WH$ production. This correlation relates very different values of the new physics scales for the fermionic and bosonic operators. In the lower panels we see how this weakens the limits on the bosonic operators $f_B$ and $f_W$ after profiling over the fermionic Wilson coefficients, and how it re-induces a correlation between them.

All of this discussion clearly defines a new challenge for global Higgs analyses once we reach Run II levels of precision: we need to include the additional operators shown in Eq.(10) [40, 41, 80–85]. As argued above, this is at least in part due to a relative enhancement of the fermionic Higgs-gauge operators through their 4-point interactions. We show the result of our global analyses in Fig. 8, both at the 68% and 95% confidence levels. As LHC observables we consider the same measurements as Fig. 4, but now combined with electroweak precision observables and including an extended set of operators. While the triple-gluon operator $\mathcal{O}_G$ and the chromo-magnetic operator $\mathcal{O}_{tG}$ appear in a global Higgs analysis, we have shown in Sec. 4 that their best limits come from dedicated studies and after considering these limits their effects on the Higgs observables will not be visible. We therefore include them in

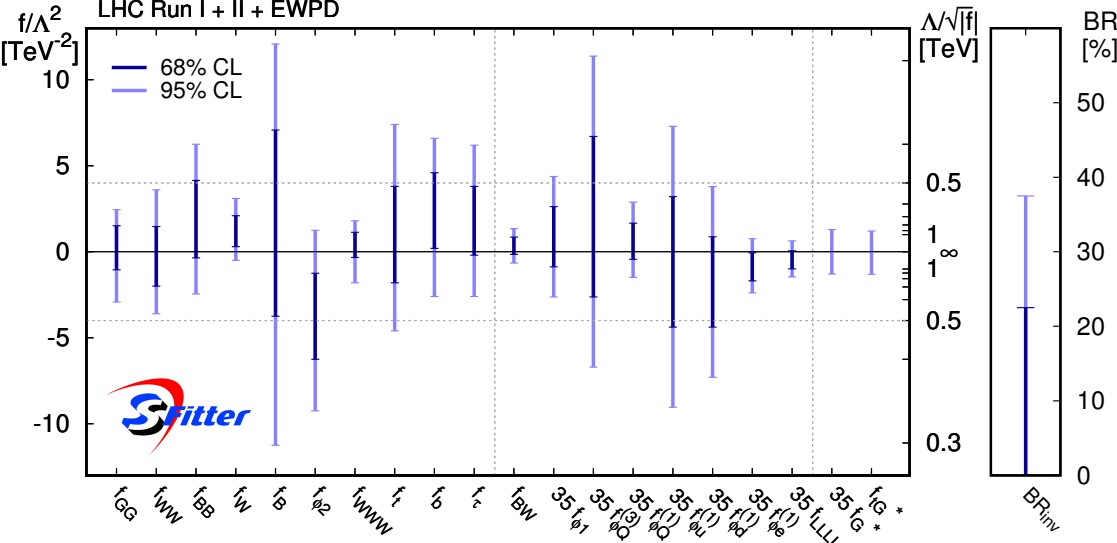

Figure 8: Allowed 68% and 95% CL ranges for individual Wilson coefficients $f_x/\Lambda^2$ from a one-dimensional profile likelihood. All results include the Run II measurements combined with electroweak precision data. We quote the best results for $\mathcal{O}_G$ [59] and $\mathcal{O}_{tG}$ [65] from non-Higgs analyses.

the SMEFT-like result shown in Fig. 8, but quote the constraints from non-Higgs analyses.

First, we see that the 68% and 95% confidence limits scale like we would expect from Gaussian uncertainties. Directly comparing the results for the bosonic operators without and with the fermionic operators we see that as expected from Fig. 7 the results on $f_B$ are roughly a factor of two weaker once we profile over the fermionic Wilson coefficients. We also see weaker limits on $f_W$ and $f_{\phi 2}$, which propagate through the entire effective Lagrangian describing the global analysis.

The constraints from our global analysis on the fermionic Higgs-gauge operators are typically a factor 10 to 100 stronger than for the bosonic operators. With $f_{\phi Q}^{(3)}$ and $f_{\phi d}^{(1)}$ the global fit also constrains operators which are relatively poorly probed by electroweak precision observables alone. These limits are in the range of $\Lambda/\sqrt{f} \approx 3$ TeV at 68% CL, indicating that LHC observables can also be especially sensitive to these operators. Again, for these results it is crucial that our global Higgs analysis covers Higgs observables and di-boson observables at the LHC, combined with electroweak precision data.

# 7 Summary

We have presented a global analysis of the LHC Run I and Run II measurements related to Higgs and di-boson measurements in the framework of an effective Lagrangian to dimension six. The increasingly strong constraints from Run II and especially the developing LHC sensitivity to anomalous gauge boson couplings to quarks require a combination of the LHC analysis with electroweak precision data. In our global Higgs and electroweak analysis we include 18 bosonic and fermionic dimension-6 operators. For two more operators we quote limits from other analyses, after confirming that they are more constraining than our Higgs analysis. Finally, we include invisible Higgs decays through their branching ratio. This set of operators defines a significant step towards a global SMEFT analysis in the LHC era and towards a global precision analysis of LHC data.

In the SFITTER framework we directly analyze ATLAS and CMS measurements rather than pre-defined pseudo-observables, include correlations for systematic and theoretical uncertainties, and exploit kinematic distributions to large momentum transfer. For LHC data alone we find that all limits from Run I are consistently improved by Run II, especially in the Yukawa sector and from the kinematic measurements of $VH$ production. At 95% CL the typical Run II limits range around $\Lambda/\sqrt{f} = 400 \dots 800$ GeV. Through new 4-point vertices fermionic Higgs-gauge operators have an anomalously large effect on associated Higgs production. This induces strong correlations between fermionic operators and $f_{B,W}$, in spite of stringent constraints from electroweak precision data. Profiling over the fermionic Wilson coefficients weakens the limits on $f_B$ by a factor two. At the same time, LHC observables allow us to constrain fermionic operators like $f_{\phi d}^{(1)}$ far beyond the reach of electroweak precision data, indicating that the interaction between the two sectors of our global fit is mutual. For several bosonic operators our analysis probes $\Lambda/\sqrt{f}$ values up to the TeV range, while the fermionic Higgs-gauge operators are consistently constrained to $5 \dots 10$ TeV.

## Acknowledgments

We are grateful to Dirk Zerwas and Michael Rauch for their continuous support of SFITTER and for many discussions related to LHC measurements. We would like to warmly thank Tatjana Lenz and Ruth Jacobs for their help with the $m_{VH}$ distribution. We would like to thank the DAAD Australia exchange program for funding our project *Precision Higgs Physics at the LHC* (57390316). AB is funded through the Graduiertenkolleg *Particle physics beyond the Standard Model* (GRK 1940). TC acknowledges generous support from the Villum Fonden and partial support by the Danish National Research Foundation (DNRF91) through the Discovery centre. The authors acknowledge support by the state of Baden-Württemberg through bwHPC and the German Research Foundation (DFG) through grant no INST 39/963-1 FUGG (bwForCluster NEMO).

## A    Constraints on fermionic operators from EWPD and the LHC

In Section 6, we have seen that the additional, mostly fermionic, operators of Eq.(10) have a non-negligible impact on the limits of the Wilson coefficients on the bosonic operators in Eq.(3). We have explicitly studied the effect of the fermionic Higgs-gauge operators on the limits on $f_B$, see also Fig. 5 and Fig. 7. The interplay between electroweak precision data and LHC observables is mutual in the sense that limits on operators constrained by EWPD also receive important contribution from LHC data. We demonstrate the impact of LHC observables on the operators in Eq.(10) in Fig. 9, where we compare the limits resulting from a fit of EWPD only with a combined fit of EWPD with LHC Run I+II observables. The limits on the Wilson coefficients, especially on those of fermionic Higgs-gauge operators, are consistently improved by the inclusion of LHC data and asymmetries of the limits are reduced. This highlights the relevance of LHC measurements for the precise determination of the couplings of gauge bosons to fermions.

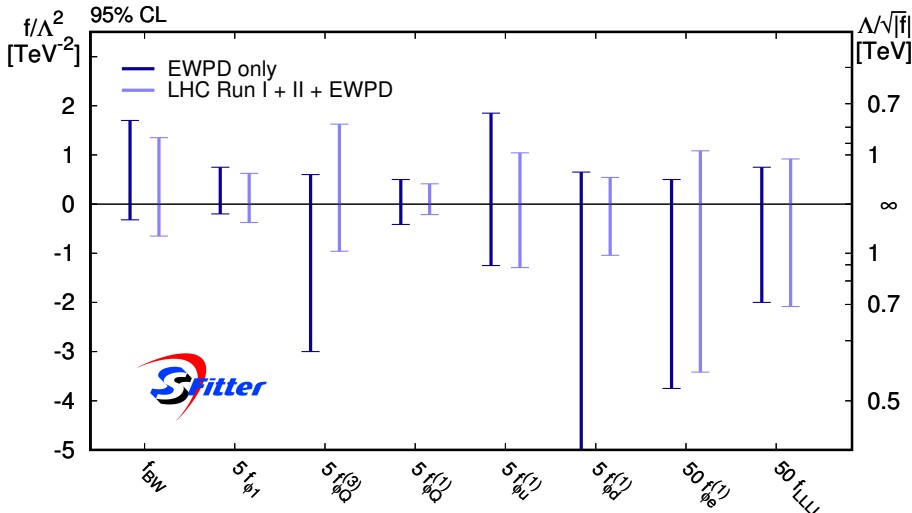

Figure 9: Allowed 95% CL ranges for individual Wilson coefficients $f_x/\Lambda^2$ of the operators given in Eq.(10) from a one-dimensional profile likelihood. We compare the constraints from electroweak precision data with a combination of EWPD and LHC Run I+II measurements. The lower limit on the Wilson coefficient $f_{\phi d}^{(1)}$ from the fit of EWPD only is $-1.2\ \mathrm{TeV}^{-2}$.

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
