# Peer review of "The Gauge-Higgs Legacy of the LHC Run II"

_SciPost Physics, doi:SciPost Phys. 6, 064 (2019)_

## Round 1 · Referee Report · Anonymous (Referee 1) · 2019-1-19

Strengths

1 Precise and careful reanalysis of experimental data.
2 Systematic approach in terms of operators

Weaknesses

1 We already had bounds, the new ones don't have a significant impact
2 Simpler approaches give most of the information

Report

The paper presents a detailed accurate updated analysis of precision bounds on extra dimensional operators, including data from the LHC collider. While experts can appreciate the systematic approach, the paper does not add much to what we already knew

Requested changes

In my opinion the paper can be published in its present form

---

## Round 1 · Referee Report · Anonymous (Referee 2) · 2019-2-6

Strengths

1) Fit with new Run2 Data 2) The authors take the opportunity to re-discuss practices that have become common in the literature (in particular they quantify the extend to which operators that enter EW observables can be neglected in the context of Higgs physics)

Weaknesses

1) Physics content not very innovative 2) Some explanations not very clear (see requested changes)

Report

To be published after adressing comments below.

Requested changes

1) The authors mention Triple gauge vertices (TGV) in section 2, before including the operators of eq 8, some of which also enter diboson pair production and play therefore a role in the extraction of TGV. I did not understand whether these effects (eq 8 in dibosons and the extraction of TGV) has been taken into account?

2) In the context of Fig.1, the third and fourth diagrams are described as providing different energy growth (it is said that the third is accompanied by a propagator, while the fourth is not). It seems to me that, when included in the amplitude, diagrams 3-4 give exactly the same contribution (the momentum in vertex 3, and the one from a SM triple gauge vertex will compensate the $p^2$ in the propagator). Moreover in the caption it is said that the scaling with $v$ is generated by the non abelian component; I don't understand this statement.

3) In the last paragraph of section 4: the authors say that the tG operator enters $tt$ production, which tests it much better than $tth$. It is not obvious to me that this dominance will always persist: it enters $tt$ with the opposite chirality (helicity in the $E\gg m_t$ limit) w.r.t the SM, so it doesn't interfere at the leading order in $E/m_t$; moreover the $tG$ operator is suppressed by an insertion of $v$ in the $tt$ process. On the other hand this operator sources the same helicity as $tth$ in the SM and it represents a contact interaction for $tth$, so that it must grow as $E^2/Lambda^2$ w.r.t. the SM. So I expect that, as the collider energy increases (as in Run 2), tth might play more and more of an important role.

4) In p.10 it is said that $h\to 4l$ can be neglected in a global analysis; are the authors referring to the differential measurement or even the branching fraction?

5) Figure 5: I find it confusing that the shaded band does not include the SM, which lies between the opposite signs of the coefficient. Also the units don't match (shouldn't it be TeV$^2$?).

6) top of p13: the authors mention that the error bars in fig 3 are not gaussian and symmetric: is there a quantitative way to relate these shapes to statements about, say, the relative size of linear and quadratic terms (I guess this is where the authors are heading)? Also, further down \emph{extremely successful} Run I measurements: is this for a statistical fluctuation?

7) Fig.6 and discussion in p14. I have similar comments as above for fig1. It doesn't seem to me that the scalings in eq 20 can be used to conclude their impact on the amplitude: longitudinal vector polarizations have power of $E/v$ that also enter and compensates for the powers of $v$ at numerator in eq 20. It seems to me that these vertices have the same size in diagrams.

Moreover: what couplings are the authors referring to in the last paragraph of p 14?

---

## Round 2 · Referee Report · Francesco Riva (Referee 3) · 2019-5-10

Report
The authors have thoroughly addressed all the comments and I recommend the paper for publication.

---

## Round 2 · Author Response

We thank the referee(s) for their very useful and valuable input.

---

## Round 2 · List of Changes

We addressed the concerns of the referee in the following way:
1 We are including fermionic operators in the study of triple gauge vertices. In addition to the sentence 'The three Wilson coefficients relevant for our analysis of di-boson production are fB,fW and fWWW, plus the operators influencing electroweak precision data discussed in Section 3.' that we already had on page 4, we added a comment in Section 3:
'The challenge is that the bosonic
operators in Eq.(7) and the fermionic operators in
Eq.(8) not only contribute to electroweak precision
physics, but also to di-boson or Higgs production at an observable
level, where they are included e.g. in our study of triple gauge vertices.'
2 We agree with the comments of the Referee and have changed the language to reflect that the amplitude does not have additional momentum enhancement over the SM.
3 We have added a sentence emphasizing the measured cross sections for tt and tth production and arguing that the sheer number of events in tt production implies a statistical advantage in measuring ftG over the tth channel. We have also referenced arXiv:1607.05330 which finds for the HL LHC the constraints from tth on f_tG will exceed those from tt in agreement with the referee.
In terms of the chiral vs helicity analysis we researched this and found it is a discussion well beyond the scope of our analysis - arXiv:1205.1065 figure 6 indicates that ttH with an insertion of O_tG sources LL+RR and LR+RL helicity combinations. Dropping the Higgs lines in Figure 6 implies tt production sources LR helicity combinations and the 5-point contact vertex also sources LR.
Looking at arXiv:1403.1790 Figure 1 which plots the hardest top pT against the fraction of the total cross section corresponding to each chirality, as the pT increases tt sources almost exclusively LL+RR (labelled LR+RL due to a different convention, naming the chirality of the antiparticle instead) while ttH sources equally both LL+RR and LR+RL. Thus we expect a decrease in tt production as the SM for high pT is dominantly LL+RR while O_tG is only LR, while for tth we expect no decrease as O_tG sources all helicity combinations.
4 We are referring to differential measurements and have changed the wording accordingly.
5 We have corrected the labels from Lambda^2 to Lambda.
We agree that the shading of the area between the curves is misleading. We are now only showing the lines for +/- a specific Wilson coefficient and state so in the caption of the Figure.
6 The relative size of the linear and quadratic terms is one, but not the only effect influencing the shape of the error bars. We have expanded the sentence to
'However, from Fig.3 we know that the error bar on $f_W$
is by no means symmetric and Gaussian due to
the relative size of the linear and quadratic terms of the EFT, the
parametrization of the theory prediction and further effects.'
By the 'extremely successful' Run I measurements, we do not mean statistical fluctuations. Instead, this is rather meant as an acknowledgement of the breadth of powerful Run I measurements the experimentalists performed, showing the right distributions up to relatively high energies. For Run II however, we are still waiting for WW measurements with an inverse luminosity of > 3.2 ifb...
We added the word 'broad' to the sentence '...still dominated by a broad set of extremely successful kinematic
measurements at Run~I in view of a global gauge analysis.' to highlight this point and make clear that we are talking about the impact on a global analysis.
7 We are referring to the coupling g' and are explicitly stating so in the text now. We have also clarified that the operator definitions include gauge couplings. The scaling of the diagrams in Figure 6 has been clarified, in particular the diagrams with propagating Zs are suppressed by the off shell propagator going as 1/(mZH^2-MZ^2)<1. The referee's comments about longitudinal vector polarizations apply to all three diagrams as they all have Zs in the final state.
1 We are including fermionic operators in the study of triple gauge vertices. In addition to the sentence 'The three Wilson coefficients relevant for our analysis of di-boson production are fB,fW and fWWW, plus the operators influencing electroweak precision data discussed in Section 3.' that we already had on page 4, we added a comment in Section 3:
'The challenge is that the bosonic
operators in Eq.(7) and the fermionic operators in
Eq.(8) not only contribute to electroweak precision
physics, but also to di-boson or Higgs production at an observable
level, where they are included e.g. in our study of triple gauge vertices.'
2 We agree with the comments of the Referee and have changed the language to reflect that the amplitude does not have additional momentum enhancement over the SM.
3 We have added a sentence emphasizing the measured cross sections for tt and tth production and arguing that the sheer number of events in tt production implies a statistical advantage in measuring ftG over the tth channel. We have also referenced arXiv:1607.05330 which finds for the HL LHC the constraints from tth on f_tG will exceed those from tt in agreement with the referee.
In terms of the chiral vs helicity analysis we researched this and found it is a discussion well beyond the scope of our analysis - arXiv:1205.1065 figure 6 indicates that ttH with an insertion of O_tG sources LL+RR and LR+RL helicity combinations. Dropping the Higgs lines in Figure 6 implies tt production sources LR helicity combinations and the 5-point contact vertex also sources LR.
Looking at arXiv:1403.1790 Figure 1 which plots the hardest top pT against the fraction of the total cross section corresponding to each chirality, as the pT increases tt sources almost exclusively LL+RR (labelled LR+RL due to a different convention, naming the chirality of the antiparticle instead) while ttH sources equally both LL+RR and LR+RL. Thus we expect a decrease in tt production as the SM for high pT is dominantly LL+RR while O_tG is only LR, while for tth we expect no decrease as O_tG sources all helicity combinations.
4 We are referring to differential measurements and have changed the wording accordingly.
5 We have corrected the labels from Lambda^2 to Lambda.
We agree that the shading of the area between the curves is misleading. We are now only showing the lines for +/- a specific Wilson coefficient and state so in the caption of the Figure.
6 The relative size of the linear and quadratic terms is one, but not the only effect influencing the shape of the error bars. We have expanded the sentence to
'However, from Fig.3 we know that the error bar on $f_W$
is by no means symmetric and Gaussian due to
the relative size of the linear and quadratic terms of the EFT, the
parametrization of the theory prediction and further effects.'
By the 'extremely successful' Run I measurements, we do not mean statistical fluctuations. Instead, this is rather meant as an acknowledgement of the breadth of powerful Run I measurements the experimentalists performed, showing the right distributions up to relatively high energies. For Run II however, we are still waiting for WW measurements with an inverse luminosity of > 3.2 ifb...
We added the word 'broad' to the sentence '...still dominated by a broad set of extremely successful kinematic
measurements at Run~I in view of a global gauge analysis.' to highlight this point and make clear that we are talking about the impact on a global analysis.
7 We are referring to the coupling g' and are explicitly stating so in the text now. We have also clarified that the operator definitions include gauge couplings. The scaling of the diagrams in Figure 6 has been clarified, in particular the diagrams with propagating Zs are suppressed by the off shell propagator going as 1/(mZH^2-MZ^2)<1. The referee's comments about longitudinal vector polarizations apply to all three diagrams as they all have Zs in the final state.

---

## Editorial Decision

published